# AIRRscape: An interactive tool for exploring B-cell receptor repertoires and antibody responses

**Eric Waltari** ⬚*, **Saba Nafees** ⬚, **Krista M. McCutcheon** ⬚ᵠ, **Joan Wong** ⬚, **John E. Pak** ⬚*

Chan Zuckerberg Biohub, San Francisco, California, United States of America

¤ Current address: Immune-Onc Therapeutics Inc, Palo Alto, California, United States of America
* eric.waltari@czbiohub.org (EW); john.pak@czbiohub.org (JEP)

**Data Availability Statement:** All AIRRscape code and data loaded for visualization as well as code for processing datasets are available on a GitHub repository at https://github.com/czbiohub/AIRRscape, and we have used Zenodo to assign a

## Abstract

The sequencing of antibody repertoires of B-cells at increasing coverage and depth has led to the identification of vast numbers of immunoglobulin heavy and light chains. However, the size and complexity of these Adaptive Immune Receptor Repertoire sequencing (AIRR-seq) datasets makes it difficult to perform exploratory analyses. To aid in data exploration, we have developed AIRRscape, an R Shiny-based interactive web browser application that enables B-cell receptor (BCR) and antibody feature discovery through comparisons among multiple repertoires. Using AIRR-seq data as input, AIRRscape starts by aggregating and sorting repertoires into interactive and explorable bins of germline V-gene, germline J-gene, and CDR3 length, providing a high-level view of the entire repertoire. Interesting subsets of repertoires can be quickly identified and selected, and then network topologies of CDR3 motifs can be generated for further exploration. Here we demonstrate AIRRscape using patient BCR repertoires and sequences of published monoclonal antibodies to investigate patterns of humoral immunity to three viral pathogens: SARS-CoV-2, HIV-1, and DENV (dengue virus). AIRRscape reveals convergent antibody sequences among datasets for all three pathogens, although HIV-1 antibody datasets display limited convergence and idiosyncratic responses. We have made AIRRscape available as a web-based Shiny application, along with code on GitHub to encourage its open development and use by immuno-informaticians, virologists, immunologists, vaccine developers, and other scientists that are interested in exploring and comparing multiple immune receptor repertoires.

## Author summary

Technological advances in next generation sequencing have allowed for broad experimental sampling of immune repertoires, providing insight into how our immune system responds to infection, vaccination, autoimmunity, and cancer. The scale of these "big data", however, make it difficult to bioinformatically extract the key sequence features that are shared across multiple repertoires. With AIRRscape, we enable large-scale immune repertoire visualization and analysis that requires no knowledge of the command line or advanced programming. By providing the community with an open-source, interactive,

DOI to the repository: 10.5281/zenodo.7080165.
The online version of AIRRscape is also available at
https://airrscape.czbiohub.org.

**Funding:** This work was supported by the Chan
Zuckerberg Biohub. The funders had no role in
study design, data collection and analysis, decision
to publish, or preparation of the manuscript.

**Competing interests:** The authors have declared
that no competing interests exist.

and user-friendly interface, we reduce the barriers to exploring immune repertoires at scale. We demonstrate the use of AIRRscape to characterize features of immune responses to viral infection that are shared across multiple repertoire datasets.

## 1. Introduction

Individual B-cell receptor (BCR) repertoires have exceptional diversity, estimated to be greater than $10^9$ in a single adult [1,2]. Improvements in sequencing capability over the past twenty years have allowed a wider sampling of BCR sequences to be uncovered, e.g. [3–5]. These adaptive immune receptor response sequence (AIRR-seq) datasets can offer a complex and information-rich glimpse into B-cell immune responses to vaccination and infection. For example, with analyses of AIRR-seq data comes an increasing recognition that convergence among BCR repertoires, sometimes termed 'public' clonotypes [1,2,6,7], is important for understanding the humoral immune response to natural infections and for improving vaccine design.

The size of AIRR-seq datasets can make it challenging to visualize the BCR sequence features of individual repertoires, let alone of multiple repertoires concurrently. Several groups have developed metrics to assess similarity among entire BCR repertoires [8,9], but few open-source tools enable facile, multi-dimensional visualization and exploration of these repertoires. An overview of existing tools and their features [10–21] is provided in Table 1. Notable examples of higher-level BCR repertoire visualizations (e.g. for features such as V-gene and J-gene usage, CDR3 length, and CDR3 amino acid sequence motifs) include Circos plots [22], radial phylogenies [23], and clouds summarizing clonotype networks [24]. While powerful in offering a global overview of AIRR-seq data, these methods are not amenable to interactive exploration, in particular to uncover and display antibody convergence.

To enable simultaneous interactive visualization of multiple BCR repertoires and in-depth data exploration, we have developed an open-source tool called AIRRscape. Analysis using AIRRscape begins with the generation of sequence feature heatmaps, which can span both individual or combined AIRR-seq datasets. Visual comparison of these heatmaps in their entirety provides a simple and intuitive global overview of differences in three coupled AIRR-seq dataset features: 1) V- and J-gene usage; 2) CDR3 length, and 3) either somatic hypermutation (SHM) or sequence read count. AIRRscape enables extraction of all sequences that fit a particular set of features from the heatmap (individual or combined datasets) into bins for finer scale local analysis. Topologies or phylogenetic networks of CDRH3 or CDRL3 amino acid motifs can be generated, on the fly, to examine clonotypes and study antibody convergence among multiple studies and patients. Recognizing the lack of consensus on how to define clonal lineages [2], we allow users to select from a range of CDR3 identity thresholds to define and visualize antibody convergence. AIRRscape uses R Shiny [25], and we have deployed it at https://airrscape.czbiohub.org. Thus, it can be easily used on any web browser without advanced programming or command-line expertise.

Here we use AIRRscape to examine collections of antibodies and BCR repertoires from three types of viral infections to address several questions. For SARS-CoV-2: how representative are published antibodies relative to healthy and COVID-19 patient repertoires, and how common are convergent antibody responses [26–28]? For HIV-1: how comparable are published anti-HIV-1 antibodies relative to anti-SARS-CoV-2 antibodies as well as to HIV-1 patient repertoires, and how common are public antibody sequences reported in [7]? For DENV: how common are the convergent antibodies reported in recent studies [29–31]? We

**Table 1. Open-source tools for comparing and visualizing BCR repertoires.**

| Tool | Summary | Features | Reference |
|------|---------|----------|-----------|
| AIRRscape | Web-based interactive tool for exploring B-cell receptor repertoires | Enables **easy input** of AIRR-seq datasets; **simultaneously visualizes** hundreds of thousands of sequences and networks of CDR3 motifs for analysis of convergence | This study |
| AncesTree | Interactive immunoglobulin lineage tree visualizer | Enables exploration of antibody clonal lineages processed following AIRR Community standards using a Java-based GUI | 10 |
| ASAP | Web server for AIRR-seq analysis pipeline | Processes and visualizes multiple repertoires starting from paired fastq files; outputs plots of somatic hypermutation, VDJ gene usage, & clonal expansion | 11 |
| BRrepertoire | Web server for large-scale statistical analyses of repertoire data | Accepts web-based inputs from IMGT output; focuses on plots of physico-chemical properties | 12 |
| immunarch | R package for analysis of T-cell receptor and B-cell receptor repertoires | Accepts multiple input data types; generates many exploratory plot types using R commands | 13 |
| immuneREF | R package for analysis of repertoire similarity on a one-to-one, one-to-many, and many-to-many scale | Compares multiple repertoires processed via AIRR Community standards; visualizations include repertoire clustering by similarity and comparison of CDR3 amino acid occurrence and VJ usage among repertoires | 14 |
| Olmsted | Dockerized application for B-cell repertoire and clonal family tree exploration | Visualizes clonal lineages after clustering and processing of AIRR-seq data in JSON format; enables interactive exploration of clonotype phylogenies and amino acid changes | 15 |
| PASA | Web server for analysis and integration of data obtained from proteomics of serum antibodies | Enables exploration of proteomics data obtained via raw mass spectrometry data files from LC-MS/MS | 16 |
| sumrep | R package for immune receptor repertoire comparison and model validation | Compares multiple repertoires and outputs multiple similarity indices; creates plots of similarity distributions | 17 |
| VDJbase | AIRR-seq genotype and haplotype database | Has interactive modules for analysis of published AIRR-seq data including haplotype, gene, and allele usage; produces reports | 18 |
| VDJServer | Free, scalable web-based pipeline for immune repertoire analysis | Processes and visualizes repertoires starting from fastq files; outputs plots of somatic hypermutation, gene usage, clonal abundance, and diversity & selection measures | 19 |
| VDJtools | Software suite for analysis of T-cell receptor repertoires | Processes and visualizes T-cell receptor repertoire datasets | 20 |
| VDJviz | Web tool for browsing and analyzing B-cell and T-cell receptor repertoires | Uses VDJtools API; has interactive plots of VJ usage, clonal expansion, & rarefaction curves; online demo allows for analysis of up to 25 samples of 10,000 clonotypes each | 21 |

show that AIRRscape provides an intuitive and explorable visualization of the convergent antibody responses that occur in patients infected with SARS-CoV-2 or DENV, as well as the idiosyncratic responses that are typical in HIV-1 due to high SHM.

## 2. Material and methods

### 2.1 Datasets

Recent efforts by the AIRR Community [32,33] have led to the organization and standardization of T-cell and B-cell repertoire analysis, enhancing the accessibility of BCR repertoire datasets. Here we utilize two AIRR-seq data repositories, iReceptor and Observed Antibody Space [34–36], to gather BCR repertoire datasets in response to three viral pathogens: SARS-CoV-2, HIV-1, and DENV, the causative agents of COVID-19, AIDS, and dengue, respectively. Datasets primarily come from studies examining patient bulk BCR repertoires [7,23,26,29,30,37–41]. We also examine repositories of known monoclonal antibodies collated from multiple antibody discovery studies: CoV-AbDab for COVID-19 [42], and for HIV/AIDS the IEDB and CATNAP databases [43,44] and a recent dataset by Yacoob et al. [45]. For dengue, we focus on a set of described plasmablast sequences from two patients [29,31]. All datasets are summarized in Table 2.

**Table 2. Datasets used in AIRRscape.** Datasets in italics are collections of antibodies.

| Dataset | Data type | Sample | Reference | Source | Number of sequences |
|---|---|---|---|---|---|
| COVID-19 | *SARS-CoV-2 mAbs* | - | 2 | CoV-AbDab database (2) | *4,306* |
| COVID-19 | patient bulk repertoire | p11 | 23 | iReceptor (34) | 9,385 |
| COVID-19 | patient bulk repertoire | 7450 | 38 | iReceptor | 15,645 |
| COVID-19 | patient bulk repertoire | galson-01 | 26 | iReceptor | 29,795 |
| COVID-19 | patient bulk repertoire | M5 | 37 | iReceptor | 18,711 |
| Dengue | *dengue mAbs* | - | 31 | (31) | *79* |
| Dengue | patient bulk repertoire | d13 | 29 | SRA | 32,495 |
| Dengue | patient bulk repertoires | 45 patients | 30 | Observed Antibody Space (35) | 198,119 |
| Healthy control | bulk repertoire | BX | 40 | SRA | 50,942 |
| HIV-1 | *HIV-1 mAbs* | - | 43 | IEDB (43) | *98* |
| HIV-1 | *HIV-1 mAbs* | - | 44 | CATNAP (44) | *441* |
| HIV-1 | *HIV-1 mAbs* | - | 45 | (45) | *83* |
| HIV-1 | patient bulk repertoire | NIH45 | 41 | iReceptor | 14,644 |
| HIV-1 | patient bulk repertoire | MT1214 | 39 | iReceptor | 33,855 |
| HIV-1 | patient bulk repertoires | 6 CAPRISA patients | 7 | Observed Antibody Space | 184,294 |

## 2.2 Data processing

BCR repertoire datasets are first downloaded either as tab-delimited files that follow AIRR Community standards or as sequence files that are partially processed, e.g. using the pRESTO module of the Immcantation suite [46]. For the repertoires in the latter category, the processing is completed using two Dockerized Change-O scripts from the Immcantation suite, 'changeo-igblast' and 'changeo-clone' [47]. The collated lists of antibodies against SARS-CoV-2 and HIV-1 (from the IEDB database only) are focused on protein sequence and do not include SHM, the % nucleotide divergence from germline V-gene, which thus needs to be calculated separately. For those antibody sequences we first run a tblastn search of the NCBI nr/nt database to find perfectly matched nucleotide sequences, and then use the Change-O scripts for processing (after checking to ensure the source is the same antibody without codon optimization). We subsequently color the SHM values of these sequences to highlight this additional step. The resulting repertoire datasets now follow the AIRR Community standards, with columns indicating the assigned V-gene (*v_call*), assigned J-gene (*j_call*), mutation from germline V-gene (*v_identity*), and assigned V(D)J junction amino acid sequence (*junction_aa*).

A custom R function is used to further process the BCR repertoire datasets before visualization with Shiny. First we calculate the level of SHM from germline V-gene using the standard column *v_identity*. We also calculate CDR3 amino acid length as defined by the international ImMunoGeneTics information system (IMGT) by removing the first and last residue in the standard column *junction_aa*, with the new value called *cdr3length_imgt*. All sequences with stop codons and sequences with CDR3 lengths that are missing, not divisible by 3, or fewer than 3 amino acids are then removed. Lastly, we create additional columns used in our tool, namely v*gf* for the assigned V-gene family, *vgf_jgene* for the V-gene family + J-gene assignment, *ncount* for the count of sequences per unique combination of vgf_jgene + cdr3length_imgt, and *shm_mean* and *shm_max* for mean or maximum SHM from its germline V-gene, respectively, per unique combination of vgf_jgene + cdr3length_imgt. This function (AIRRscapeprocess) is embedded in the *Import Data* tab of the AIRRscape tool to allow researchers to modify and combine user-supplied datasets.

To visualize multiple repertoires simultaneously, sets of antibodies and/or repertoires are combined in two possible configurations. The first option visualizes a set as separate faceted

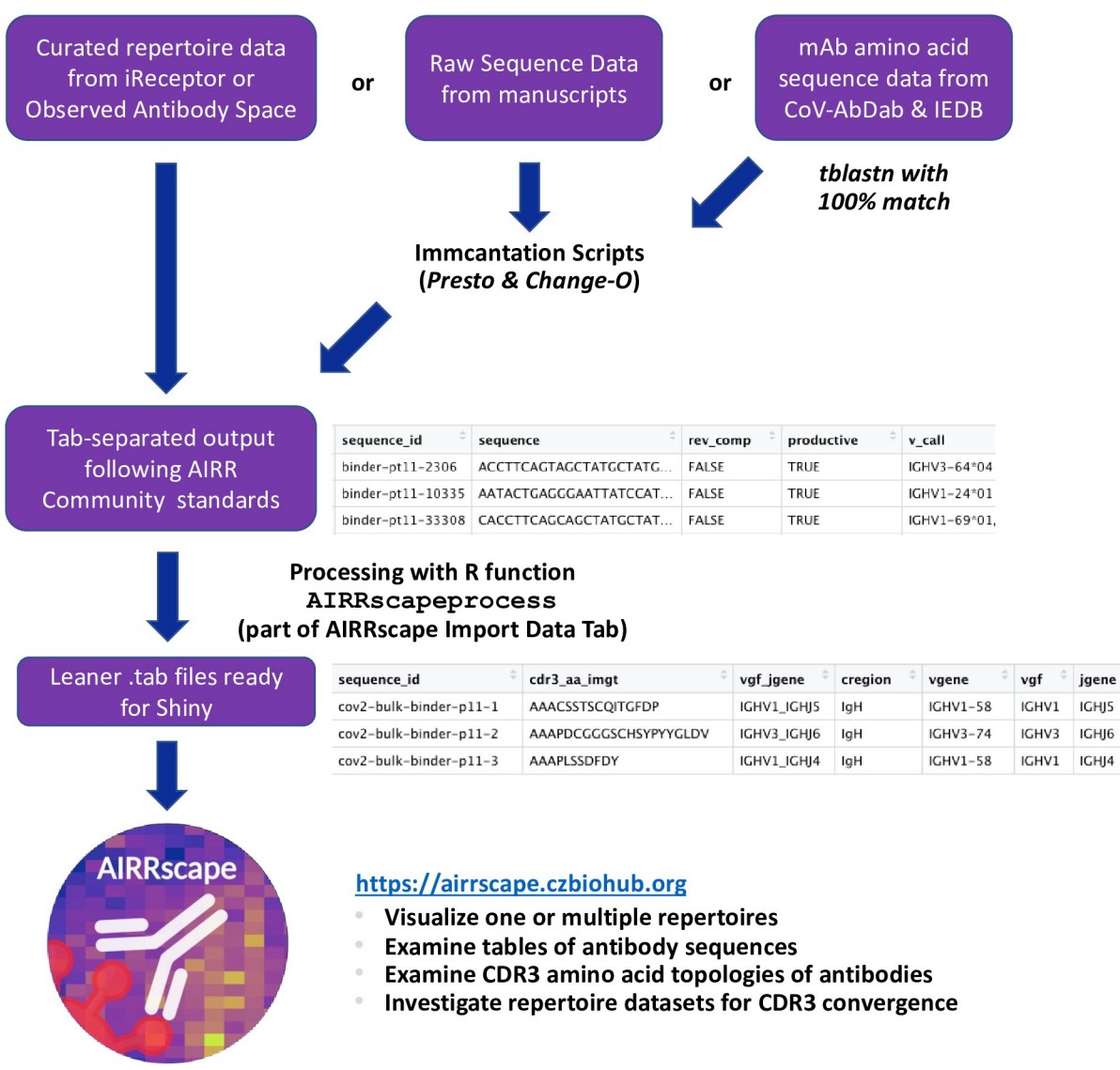

**Fig 1. Workflow of repertoire data retrieval and processing.**

panels for individual repertoires (see 2.3). The second option combines these repertoires into a single panel, to enable searches of antibody convergence. Due to their broader importance, only heavy chain antibody sequences are visualized in each of these combined repertoires [48,49]. For all of the repertoire-based visualizations, we compact the individual repertoires by collapsing all sequences with the same v*gf_jgene* + *cdr3length_imgt* bin assignments and identical CDR3 amino acid motifs. In datasets containing multiple individuals, we compact only individual repertoires before combining them. A depiction of the workflow of dataset processing is summarized in Fig 1.

## 2.3 AIRRscape application

AIRRscape is developed as an interactive web application using R Shiny. We use AIRRscape to visualize the main features of BCR repertoire datasets from processed tab-delimited files. To begin, in the *Import Data* tab researchers have the option to import, convert and combine up

to six separate BCR repertoires following AIRR Community standards (as .tsv or .tab files, no metadata required), which are downloaded in the two configurations after combining (see 2.2). Next, in the *AIRRscape* tab researchers can explore the user-supplied or pre-loaded datasets as a generated heatmap with either multiple panels of individual BCR repertoires (Fig 2A) or a single panel combining multiple repertoires (Fig 2B). The x-axis of the heatmap shows antibodies binned according to their assigned V-gene family + J-gene germline, while the y-axis shows CDR3 length. Left of the *Import Data* and *AIRRscape* tabs, the user can choose datasets to visualize one of three parameters for coloring the bins: 1) by percent of overall total antibody sequences, 2) by average SHM of the bin, or 3) by maximum SHM of the bin. The user can then hover over the paneled heatmaps to view an interactive popup displaying a bin's attributes (Fig 2B). Clicking on a single bin or selecting multiple bins using a drawn rectangle produces a table of antibodies and their features (Fig 3). Within the table, users can query by or focus on any of the antibody features loaded in the datasets. Selected antibodies can be downloaded or further explored via their topologies (i.e. their phylogenetic relationship, with the caveat that convergent antibodies are not clonally related).

Below the table of selected antibodies, AIRRscape displays interactive topologies of CDR3 amino acid sequence similarity (Fig 3). Binning antibodies both by germline assignment as well as CDR3 length enables examination of these CDR3 motifs in a phylogenetic framework, since the CDR3 motifs will be 'aligned' when constrained to bins with a given CDR3 length. The labels of the topology tips conveniently display antibody names, assigned V-genes, and CDR3 amino acid sequences. The major advantage of this approach is that thousands of antibody CDR3 sequences can be visualized via a web browser for quick and easy exploration by researchers without command-line expertise.

AIRRscape allows for multiple options to create CDR3 topologies. First, a user may select a set of antibodies from the table for constructing their topology via either the neighbor-joining or parsimony tree building methods using the Phangorn package [50,51]. Alternatively, a user can select a single antibody sequence of interest to identify all similar sequences in that bin within a chosen amino acid identity threshold. AIRRscape will then display a parsimony topology of these nearest sequences. The options for identity threshold are 50%, 70% used in [7], 80% used in [6], and 100% used in [1]. Topology tip colors are unique to each dataset. Lastly, the user has the ability to change both the height and width of the topology. Height adjustment allows the user to more easily visualize a small topology or examine a large topology (up to 500 sequences) across more than one visible page; adjusting the width allows the user to see the full tip label when there are long branch lengths.

Two custom R scripts, *airrscape_preprocessing*.R and *airrscape_processing*.R, are included to document the data processing and to manually convert files that follow the aforementioned AIRR Community standards for loading into AIRRscape. Both the pre-processing scripts and the R Shiny code for AIRRscape are available on GitHub at https://github.com/czbiohub/AIRRscape. AIRRscape can be run locally using RStudio within a browser, or it can be used on the web at https://airrscape.czbiohub.org.

## 3. Results

To illustrate the utility of AIRRscape, we used it to explore the adaptive BCR responses to three viral pathogens: SARS-CoV-2, HIV-1 and DENV.

### 3.1 anti-SARS-CoV-2 antibody datasets share characteristics and convergent motifs

We first examined the CoV-AbDab database of validated monoclonal anti-SARS-CoV-2 antibodies and compared them to published bulk BCR repertoires of four COVID-19 patients, as

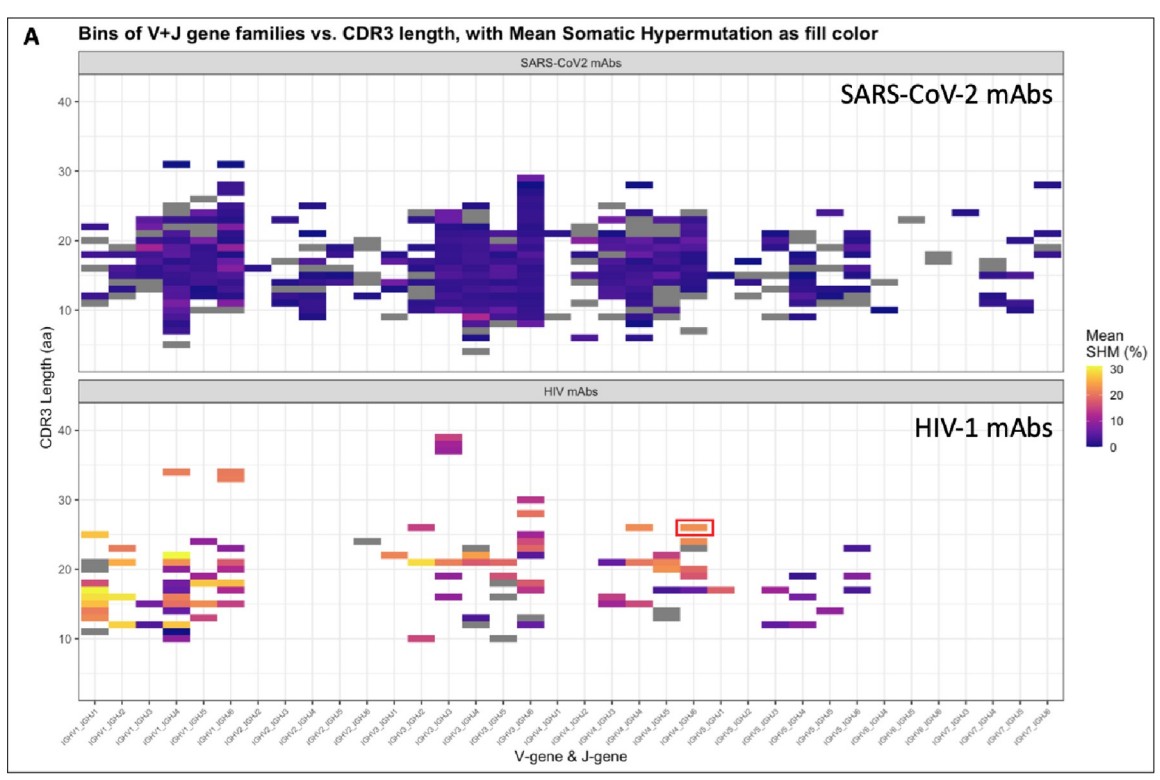

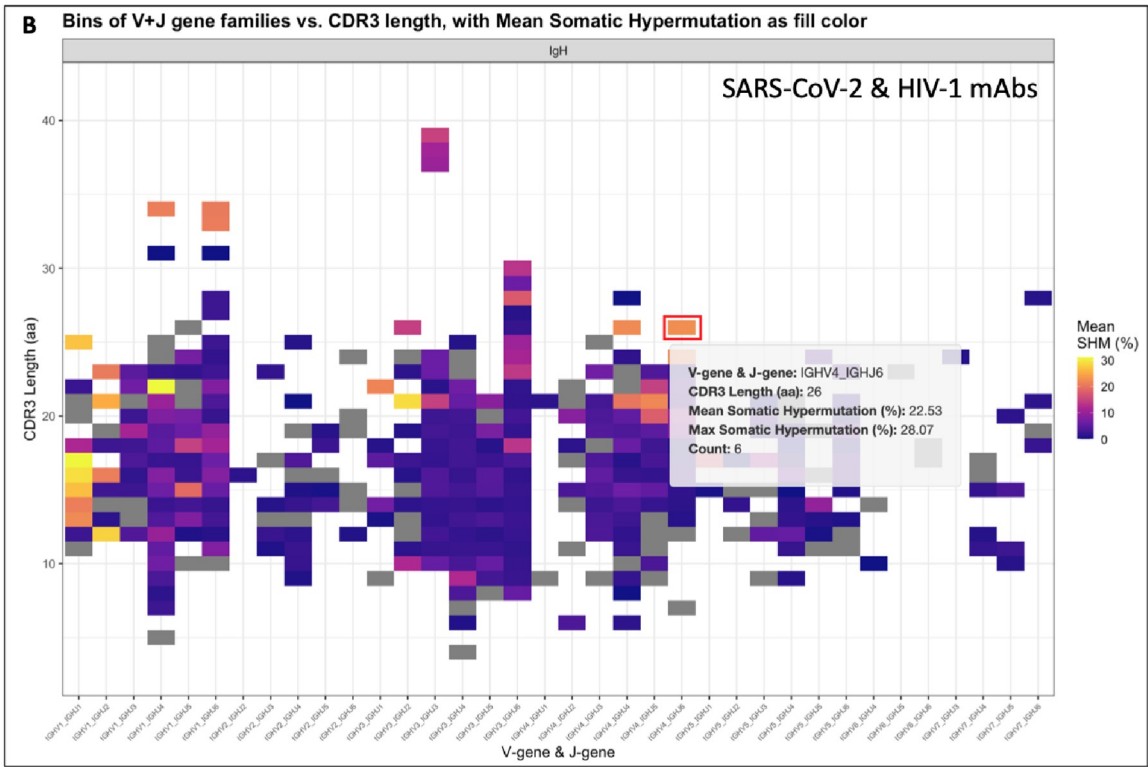

**Fig 2. AIRRscape visualization of immune repertoires.** Heatmaps comparing characteristics of (A) separate and (B) combined datasets for anti-SARS-CoV-2 and anti-HIV-1 antibodies.

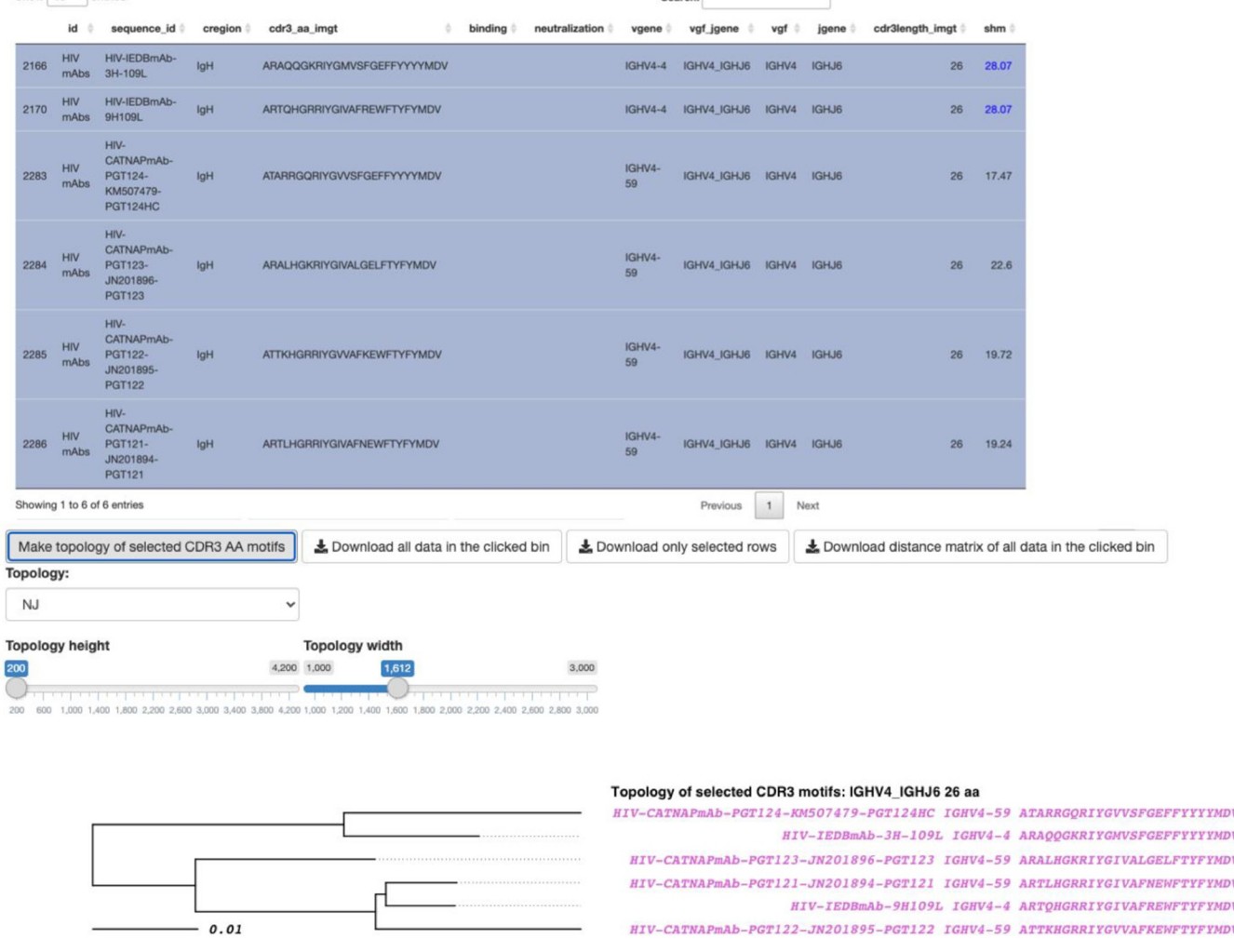

**Fig 3. AIRRscape interface showing antibodies and CDR3 amino acid topology of selected antibody bin.** Selected bin is highlighted in the red box of Fig 2B. SHM values in blue are calculated after an additional tblastn search of the NCBI nr/nt database.

well as to a healthy control bulk BCR repertoire. We visualized 2,153 paired antibody sequences from the CoV-AbDab dataset and a single representative repertoire from each study of COVID-19 patient bulk repertoires ([23,26,37,38], Table 2). After removing duplicate sequences (as described in 2.2), we compared 73,536 COVID-19 patient antibody CDRH3 sequences and 50,942 healthy control sequences (Fig 4).

We first asked how representative the anti-SARS-CoV-2 antibodies are relative to patient repertoires and relative to a healthy control repertoire, based on the heatmap visualizations in AIRRscape. This visual analysis suggested that the overall pattern of heavy chain V-gene family + J-gene pairings versus CDRH3 length distribution appears similar among the CoV-AbDab dataset, the four patient repertoires, and the healthy control repertoire (Fig 4). The most common V+J gene family assignments in all datasets were IGHV3+IGHJ4, IGHV3+IGHJ6, IGHV4+IGHJ4, IGHV4+IGHJ6, IGHV1+IGHJ4 and IGHV1+IGHJ6. A notable visual difference between the datasets was the presence or absence of sequences assigned to IGHV7 (Fig 4), which is not unexpected given that this gene 'family' consists of a single functional V-gene that does not occur in all individuals [1]. SHM levels among the anti-SARS-CoV-2 antibodies

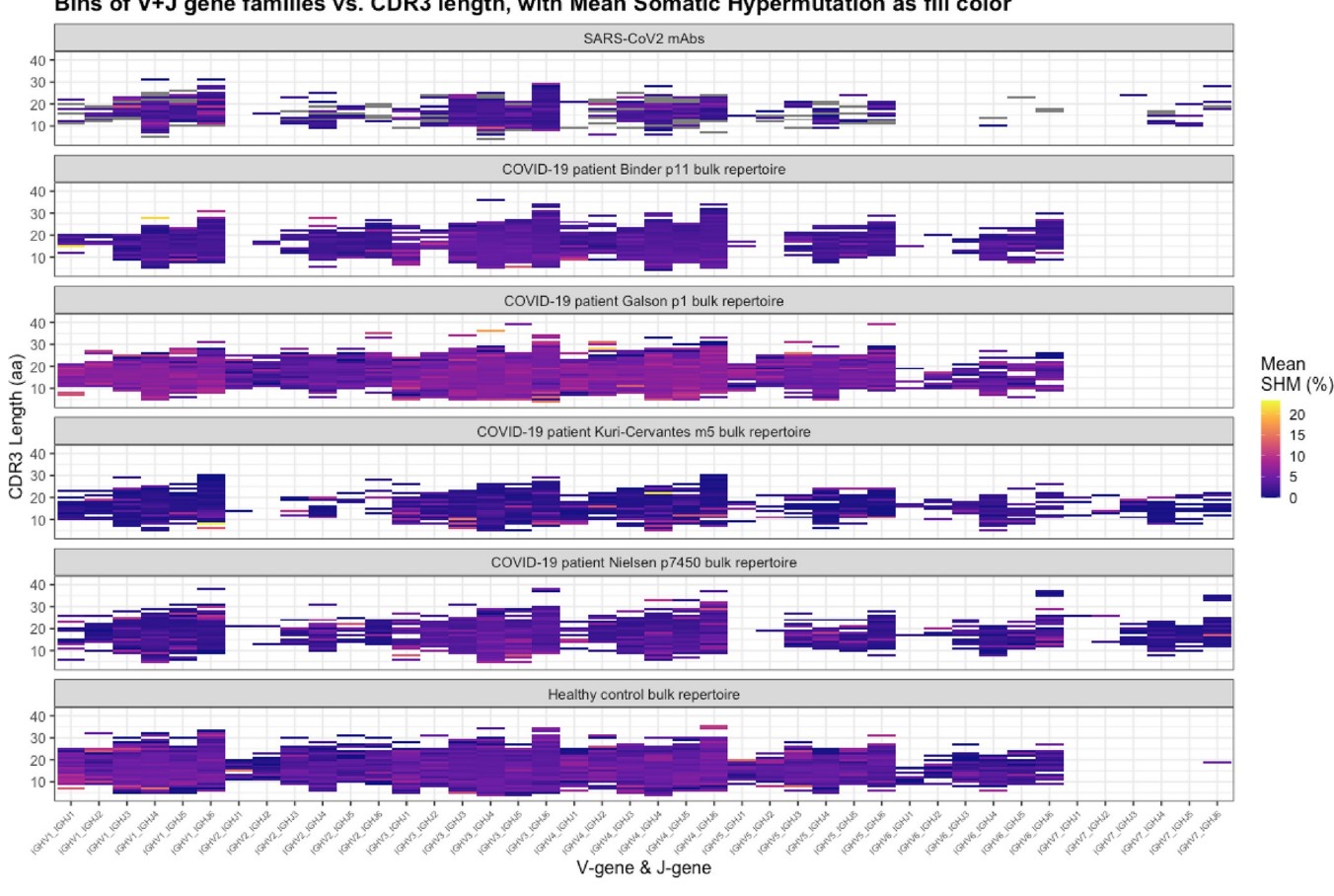

**Fig 4. AIRRscape heatmaps comparing anti-SARS-CoV-2 antibodies, bulk BCR repertoires of four COVID-19 patients, and a healthy control bulk BCR repertoire.**

(mean 2.3%) were lower than the four patient repertoires (overall mean 3.7%) and the healthy control repertoire (mean 3.3%), although the patient from the Galson et al. study had noticeably higher SHM levels (mean 5.5%). These data suggest that a majority of neutralizing antibodies against SARS-CoV-2 do not diverge greatly from germline and therefore may be more easily elicited among patients.

Next we used AIRRscape to visualize convergent clonotypes among anti-SARS-CoV-2 antibodies and COVID-19 patient bulk BCR repertoires. Visualizations were created based on their v*gf_jgene* + *cdr3length_imgt* bin assignments, where a bin *x_y_z* represents IGHV*x* germline assignment + IGHJ*y* germline assignment + CDR3 length of *z* aa. As use cases, we focused on three recently reported convergences, visualized as four bins: both *3_4_11* and *3_6_11* for an IGHV3-53-based cluster as reported by Yuan et al. [28] and Galson et al. [26]; *3_6_14* for a large cluster found by Galson et al. [26]; and *1_3_16* for a cluster reported by Robbiani et al. [27]. Within each bin, we selected one published monoclonal antibody and examined the topology of similar CDRH3 motifs among all the COVID-19 datasets, using the 80% aa sequence identity threshold selected by Soto et al. [6]. As expected, convergence across all four bins was observed (Fig 5 and Figs A-C in S1 Data), with each containing 12–30 CDRH3 motifs from the CoV-AbDab dataset; these motifs were found across seven or more unique studies. Three of the bins show convergent motifs among multiple patient repertoires (Fig 5 and Figs A and C in S1 Data), with two also showing convergence among three of the four patient

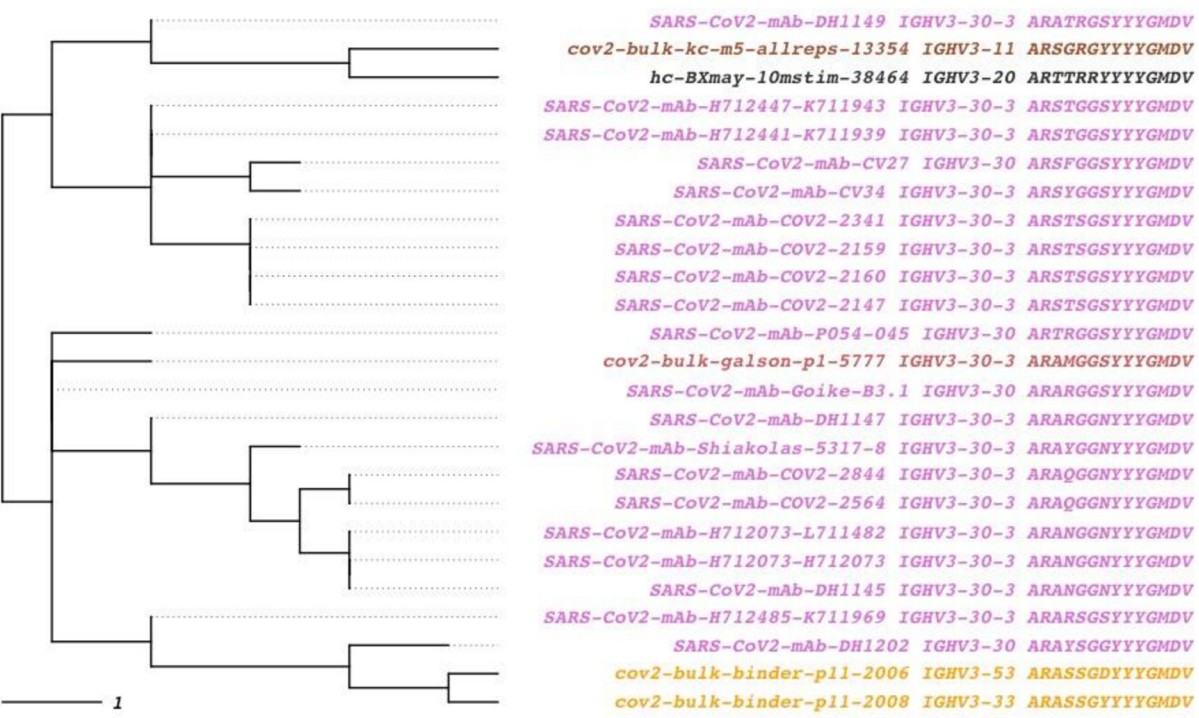

**Fig 5. SARS-CoV-2 convergent clonotypes to mAb DH1149 in the 3_6_14 bin.** An 80% identity threshold is used to calculate convergence. Tips are colored by dataset source. Purple tips are published anti-COVID-19 antibodies from 7 different studies, dark gray tips are antibody sequences from a healthy donor BCR repertoire, and orange through brown shaded tips are antibody sequences from COVID-19 patient BCR repertoires.

repertoires. Furthermore, two of the bins show convergence to the healthy control repertoire (Fig 5 and Fig A in S1 Data). The three convergent motifs seen in multiple COVID-19 patient repertoires were also found in HIV-1 and dengue patient repertoires (see 3.4). While SHM is not included in the CDRH3 topologies, examination of the convergent sequences indicated that they all show low SHM (0–4%).

## 3.2 anti-HIV-1 antibody datasets display idiosyncrasies

We collated anti-HIV-1 monoclonal antibodies from two databases and one study [43–45], and compared this set to published bulk BCR repertoires of eight HIV-1 patients. From the databases we visualized 358 sequences. For the HIV-1 patient bulk BCR repertoires, 33,855 antibody sequences from patient MT1214 [39], 14,644 from patient NIH45 [41], and 184,294 from six CAPRISA patients [7] were visualized after removing duplicate sequences (see 2.2; Fig D in S1 Data). Although patient NIH45 was only sequenced for IGHV1 genes, this partial 'repertoire' was included because it is the original source of the broadly neutralizing antibody VRC01 [41].

The heatmap visualizations in AIRRscape were used to compare published anti-HIV-1 antibodies to anti-SARS-CoV-2 antibodies and to two HIV-1 patient repertoires. While the number of anti-HIV-1 antibodies was sparse relative to that of anti-SARS-CoV-2 antibodies, they appeared to have similar heavy chain V-gene family + J-gene pairings (Fig 2A). There were noticeable differences in both SHM and CDRH3 length, consistent with previous anti-HIV-1

antibody repertoire characterizations [52,53]. The anti-HIV-1 antibody dataset showed higher SHM (mean 12.6% vs. 2.3%, Fig 2A) and longer CDRH3 lengths (20.4 vs. 15.8 aa) than the CoV-AbDab dataset. Comparing published anti-HIV-1 antibodies with the HIV-1 patient repertoires also revealed higher SHM (mean 12.6% vs. 5.0%) and longer CDRH3 lengths (mean 20.4 vs. 16.0 aa), which contrasts with the COVID-19 results in 3.1 (Fig D in S1 Data) and indicates strong diversifying selection away from unmutated, naive antibodies.

We used AIRRscape to visualize convergent clonotypes among anti-HIV-1 antibodies and HIV-1 patient bulk BCR repertoires, focusing on ten recently reported convergences from Setliff et al. [7], visualized as eight bins, as well as the inferred VRC01 germline CDR3 motif [41]. As per 3.1, we selected one published monoclonal antibody within each bin and examined the topology of similar CDRH3 motifs among all the HIV-1 datasets, this time using a lower 70% aa sequence identity threshold and requiring identical V-gene assignment, following Setliff et al. [7]. Four of the bins showed limited convergence among HIV-1 datasets (Fig 6 and Fig E in S1 Data), with motifs shared among two, three, or four patient repertoires. Despite including two antibody datasets from studies on the VRC01 antibody lineage, we notably could not find any convergence to the VRC01 germline CDR3 motif among the HIV-1 datasets. The complex evolution from germline and mutational requirements of neutralizing HIV-1 antibodies are consistent with the rarity of success in naturally achieving humoral neutralization in infected individuals and with challenges encountered in vaccine development.

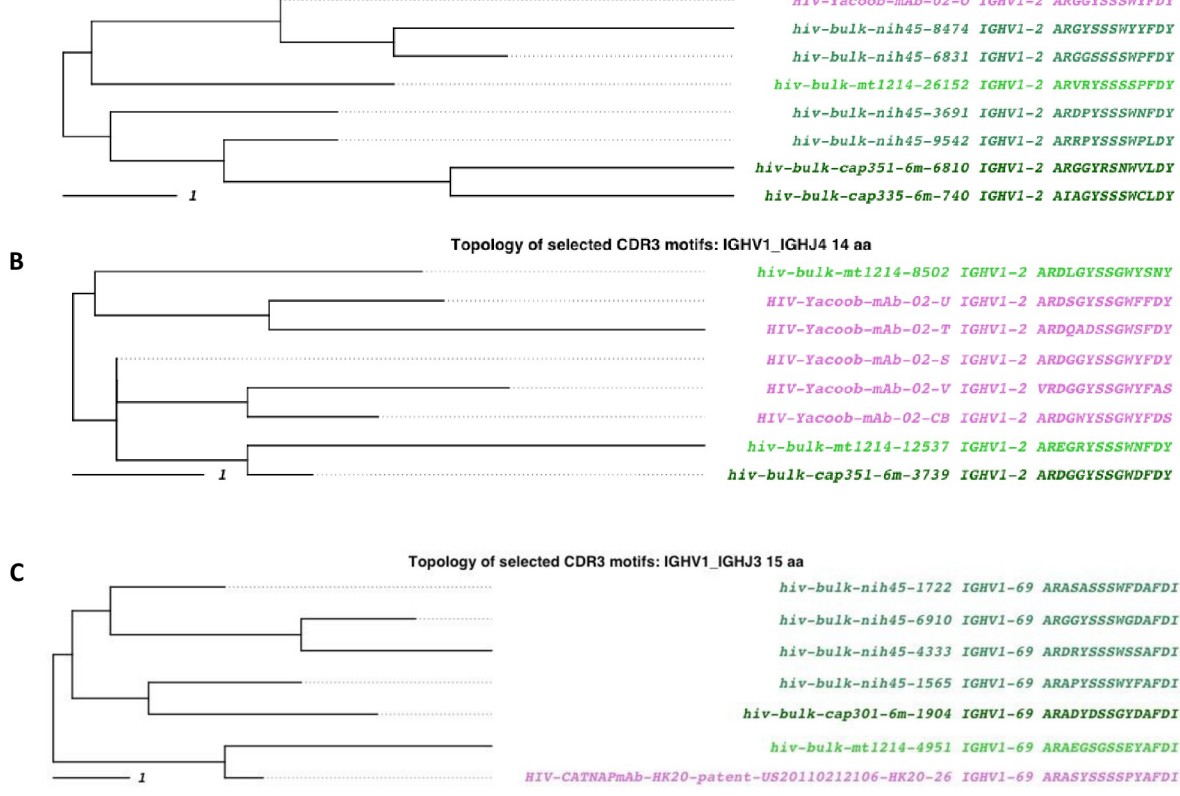

**Fig 6. HIV-1 convergent clonotypes to antibodies from Setliff et al. (2018; Fig 4).** (A) Convergent clonotypes to mAb 02-o in the 1_4_13 bin. (B) Convergent clonotypes to mAb 02-s in the 1_4_14 bin. (C) Convergent clonotypes to mAb HK20 in the 1_3_15 bin. A 70% identity threshold is used to calculate convergence. Tips are colored by dataset source. Purple tips are published anti-COVID-19 antibodies, and green shaded tips are antibody sequences from HIV-1 patient BCR repertoires.

### 3.3 anti-dengue antibody datasets share convergent motifs

AIRRscape was used to visualize both reported and potentially undiscovered convergence among the collated dengue datasets. We searched for convergence between 38 unique antibody sequences from plasmablasts isolated from two Colombian dengue patients and bulk BCR repertoires of dengue patients from Colombian and Nicaraguan cohorts. Patient bulk BCR repertoires consist of one Colombian patient (d13; [29]) and a collection of 45 Nicaraguan patient repertoires [30]. After removing duplicates, we compared 32,495 antibody sequences from Patient d13 and 198,119 from the Nicaraguan cohort (Fig F in S1 Data).

Evidence of convergent antibody motifs was previously found in independent studies of the Colombian (n = 2) and Nicaraguan (n = 45) patients. Plasmablast sequences in the Colombian patients were reported to cluster into 15 clonal families, with one clonal family (CF1) found in both patients [31]. Similarly, analysis of the Nicaraguan cohort found six CDRH3 motifs common to multiple patients [30].

In our analysis, we began with each of the 15 plasmablast clonal families, and searched for similar CDRH3 motifs among the full dengue dataset, using the 80% aa sequence identity threshold as per Soto et al. [6]. Five of the 15 clonal families showed convergence across both Colombian and Nicaraguan cohorts (Table 3 and Fig 7 and Figs G-I in S1 Data). Of these five, we identified CF1 from the Colombian cohort as the most prevalent clonal family in the Nicaraguan dataset, occurring in 16 donors (Table 3 and Fig 7). We also found a previously unreported instance of convergence in the Colombian dataset. CDRH3 motifs from the CF6 clonal family identified in patient d20 were also found in the patient d13 bulk BCR repertoire, as well as in two patient repertoires from the Nicaraguan bulk dataset (Table 3 and Fig G in S1 Data).

The six common CDRH3 motifs reported in the Nicaraguan dengue patients were also explored using AIRRscape. These six motifs group into two clusters by amino acid similarity. We found that the first cluster is common among the Nicaraguan dengue cohort, occurring in 17 patients, but not seen in the Colombian dataset. Notably, we found that the second cluster was convergent with the CF1 clonal family ([31]; Fig I in S1 Data), demonstrating its frequent occurrence. These data demonstrate how repertoire convergence in populations can occur at different geographic levels, some more restricted than others. This could be due to regional

**Table 3. Dengue convergence to plasmablast clonal families.**

| Clonal family | V_J_CDR3 length | Plasmablast donor | Colombian p13 Bulk matches? | Nicaraguan cohort matches? | Nicaraguan sequence matches | Nicaraguan donors with match |
|---|---|---|---|---|---|---|
| CF1 | 4_5_10 | p13, p20 | yes | yes | 41 | **16** |
| CF2 | 1_4_15 | p13 | yes | no | | |
| CF3 | 1_5_16 | p13 | yes | no | | |
| CF4 | 1_4_16 | p20 | no | no | | |
| CF5 | 1_3_17 | p13 | yes | no | | |
| CF6 | 1_4_13 | p20 | **yes** | yes | 2 | **2** |
| CF7 | 1_5_16 | p13 | yes | yes | 24 | **5** |
| CF8 | 1_5_17 | p13 | yes | yes | 1 | **1** |
| CF9 | 1_5_20 | p13 | yes | yes | 7 | **3** |
| CF10 | 3_5_14 | p13 | yes | no | | |
| CF11 | 3_6_23 | p13 | yes | no | | |
| CF12 | 3_4_17 | p20 | no | no | | |
| CF13 | 4_5_18 | p13 | yes | no | | |
| CF14 | 4_4_15 | p20 | no | no | | |
| CF15 | 4_5_23 | p13 | yes | no | | |

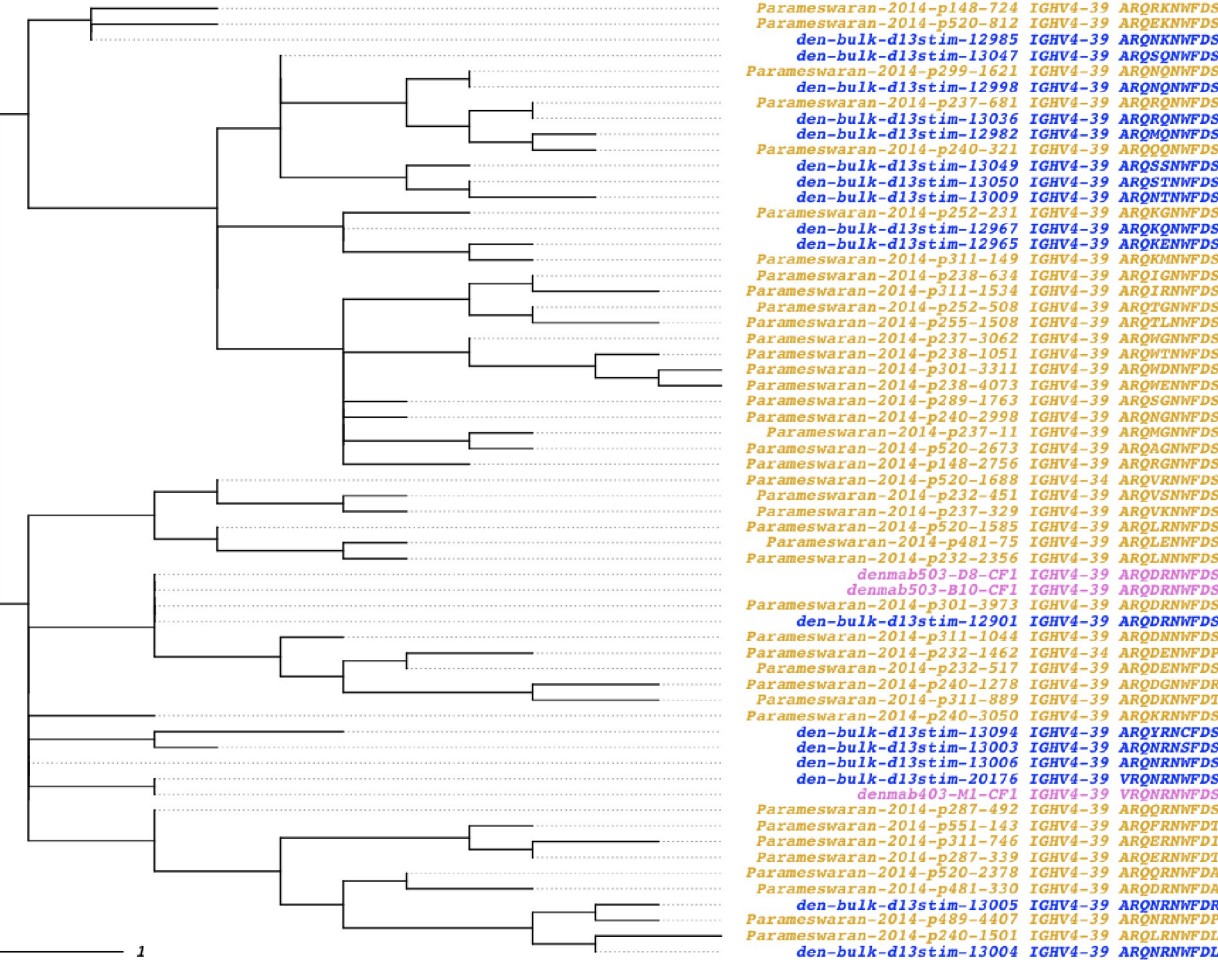

**Fig 7. Dengue convergent clonotypes to CF1 (Zanini et al. 2018).** An 80% identity threshold is used to calculate convergence. Tips are colored by dataset source. Purple tips are plasmablast sequences reported by Zanini et al. (2018) isolated from two Colombian patients (d13 and d20), blue tips are antibody sequences from the BCR repertoire of patient d13, and gold tips are antibody sequences from a cohort of Nicaraguan patient BCR repertoires.

differences or similarities in factors such as infection or vaccine history, genetics, and environment.

### 3.4 SARS-CoV-2, HIV-1, and DENV datasets share a limited set of motifs

We examined the SARS-CoV-2, HIV-1, and DENV antibody datasets together, to detect whether the reported convergent motifs are also found across diverse patient repertoires. Among the anti-SARS-CoV-2 antibody datasets, three convergent motifs seen in multiple COVID-19 patient repertoires were also found in both HIV-1 and dengue patient repertoires (Fig 8 and Figs J and K in S1 Data). The three convergent motifs, found in the 3_6_14, 3_4_11 and 3_6_11 bins, were found in 1, 3, and 2 HIV-1 patient repertoires and 6, 8, and 9 dengue patient repertoires, respectively. However, convergence with exact V-gene assignment was less common, found only in the two bins representing the IGHV3-53-based cluster (see 3.1; Figs J and K in S1 Data).

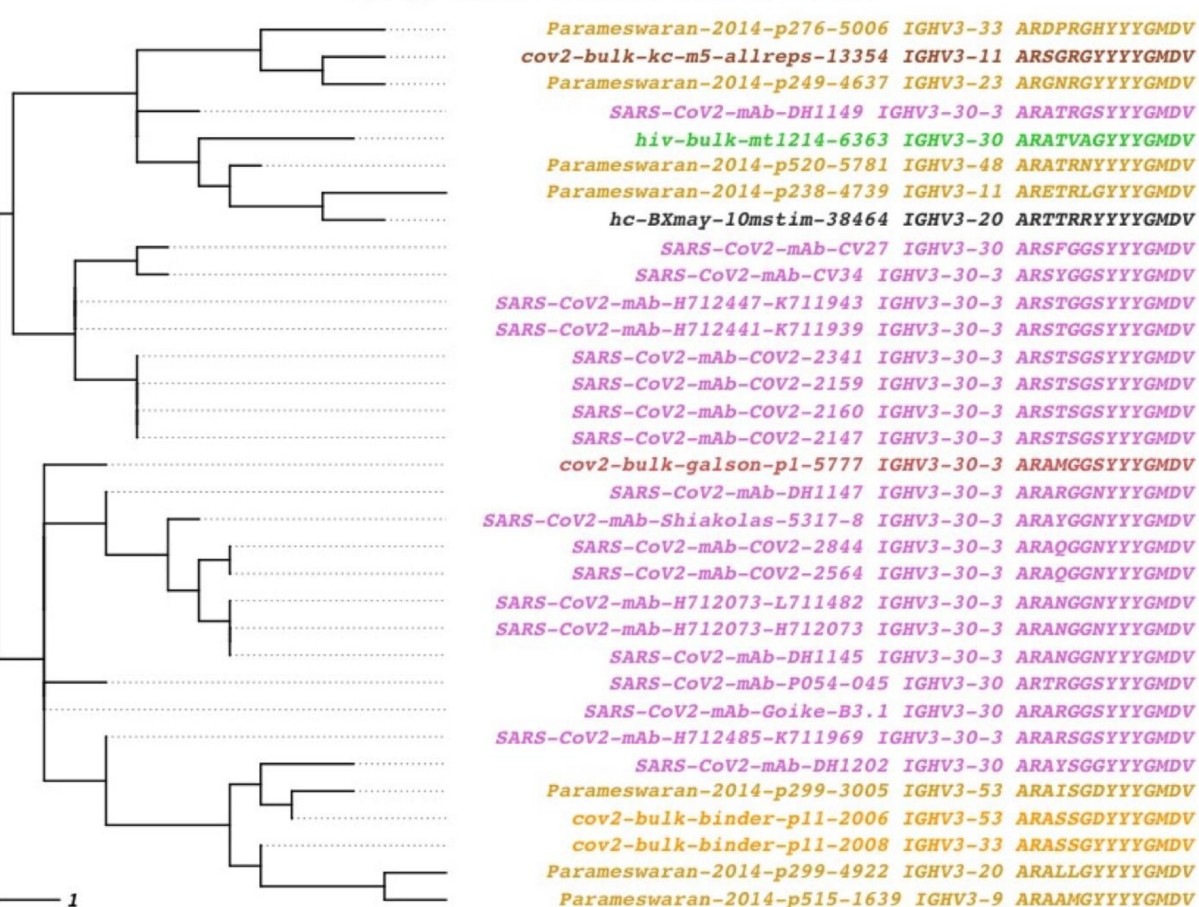

**Fig 8. SARS-CoV-2, HIV-1, & dengue convergent clonotypes to anti-SARS-CoV-2 mAb DH1149 in the 3_6_14 bin.** An 80% identity threshold is used to calculate convergence. Tips are colored by dataset source. Purple tips are published anti-COVID-19 antibodies from 7 different studies, dark gray tips are antibody sequences from a healthy donor BCR repertoire, and orange through brown shaded tips are antibody sequences from COVID-19 patient BCR repertoires. Green shaded tips are antibody sequences from HIV-1 patient BCR repertoires. Gold tips are antibody sequences from a cohort of Nicaraguan dengue patient BCR repertoires.

Among the anti-HIV-1 and anti-dengue antibody datasets, convergence across other patient repertoires was less common. Of the reported HIV-1 convergent motifs, we found no convergence among COVID-19 patient repertoires and found only one motif with convergence in dengue patient repertoires (Fig L in S1 Data). Of the five clonal families showing convergence across dengue patient cohorts, we found no convergence among HIV-1 patient repertoires, and only one motif with convergence in a single COVID-19 patient repertoire (Fig M in S1 Data). These data suggest that public antibody motifs are rare but findable using AIRRscape. However, additional knowledge of disease history in such datasets is needed before drawing conclusions. For example, it is not known whether any of the patient repertoires with such convergence have multiple viral infections.

## 4. Discussion

The importance of understanding BCR repertoires has gained increasing appreciation, especially as a result of the current COVID-19 pandemic. Along with rapidly advancing sequencing capabilities, this has led to the study and publication of dozens of individual BCR repertoires,

particularly from patients infected by well-studied infectious pathogens as well as from healthy controls, e.g. [1,6,23,26,27,37,41]. As with other examples of 'big data', there are many challenges in managing these datasets. In response, the AIRR Community has made significant efforts to standardize and curate BCR repertoires [32–34]. However, visualization and comparative analysis of these repertoires remains a challenge. To begin to address this gap we developed AIRRscape, a tool that leverages AIRR Community standards to visualize multiple BCR repertoires quickly and simultaneously. AIRRscape provides a high-level view of large datasets that allows researchers to investigate differences in commonly used repertoire characteristics, such as V-gene usage, J-gene usage, CDR3 length, and somatic hypermutation.

Recent studies have found measurable convergence between individual BCR repertoires, particularly among individuals infected with a common antigen [2]. Such convergence has great implications for understanding immune responses to antigens and informing vaccine design. Therefore, a major aim of AIRRscape is to visualize related antibody sequences, particularly from different repertoires, using the commonly accepted characteristics of convergence, V-gene assignment, J-gene assignment, and CDR3 motif. We use phylogenetic methods to more easily visualize and understand differences among the motifs, keeping in mind that the visualizations are topologies of CDR3 motifs and not necessarily phylogenetic trees indicating common ancestry, e.g. when multiple individual repertoires are compared. The characteristics used to find convergence are the same as those used to define clonotypes, or clonal clusters of antibodies within individuals. Given that no consensus on clonotype definitions has been reached, with debate largely centered around thresholds of sequence identity in the CDR3 motif, AIRRscape does not require a priori clustering of clonotypes and creates topologies based on CDR3 amino acid sequences while being agnostic with respect to thresholds.

We validated the utility of AIRRscape by exploring datasets of both antibodies and patient bulk BCR repertoires for three viral pathogens: SARS-CoV-2, HIV-1, and DENV. Among the COVID-19 datasets, we first visually confirmed that the set of known anti-SARS-CoV-2 antibodies is broadly similar both to a healthy BCR repertoire and to a collection of COVID-19 patient repertoires, as measured by heavy chain V-gene family + J-gene usage, CDRH3 length distribution, and relatively low levels of SHM from germline. This is consistent with studies made in the first months of the COVID-19 pandemic that concluded that anti-SARS-CoV-2 antibodies could likely be induced by vaccines [54]. We then used AIRRscape to examine convergence among anti-SARS-CoV-2 antibodies and COVID-19 patient BCR repertoires as reported by multiple studies [26–28]. COVID-19 convergent motifs are indeed present in currently known anti-SARS-CoV-2 antibodies as well as in four COVID-19 patient bulk BCR repertoires. COVID-19 convergent motifs were also identified among repertoires of healthy controls and dengue patients, which would not be expected to contain antibodies against SARS-CoV-2, but could be explained by the relative proximity of the motifs to germline sequences and/or to infection with other coronaviruses. Convergence of neutralizing antibody sequences among multiple COVID-19 repertoires is a strong indicator of similarity in SARS-CoV-2 immune responses and suggests that vaccines eliciting these antibodies will be broadly effective [27,55]. The discovery that some of these COVID-19 convergent motifs are public (i.e. also identified in our dengue, HIV-1, and healthy control samples) suggests that there is a general pre-existing foundation in populations towards COVID-19 protection. While a primed immune repertoire bodes well for robust vaccine responses, laboratory experiments are needed to determine whether these sequences are functional against SARS-CoV-2. Repertoires of patients infected by different SARS-CoV-2 variants could be explored using AIRRscape to find potential convergence among these datasets. Such convergence would indicate antibody motifs with potential to neutralize the range of known SARS-CoV-2 variants, which could be prioritized for reverse vaccinology research or development.

In contrast to the COVID-19 datasets, the HIV-1 datasets are mostly idiosyncratic but do show limited convergence. Using AIRRscape, we analyzed anti-HIV-1 antibodies and eight HIV-1 patient bulk BCR repertoires. The heatmaps show high SHM, a common feature of neutralizing antibodies against HIV-1. We found limited convergence among the HIV-1 datasets and no convergence to the VRC01 germline CDR3 motif [41], even using a more permissive CDR3 sequence identity threshold. While the limited convergence appears at odds with the reporting of VRC01-like antibodies in multiple HIV-1 patients [7,56], the VRC01 class varies in CDRH3 similarity and is instead defined by 20–35% SHM in the germline VH1-2 gene and a 5 aa CDRL3 length [57,58]. These idiosyncratic characteristics underscore the difficulty in vaccine development for HIV-1.

Examination of dengue datasets revealed evidence of convergent antibody motifs among two patient cohorts from Colombia and Nicaragua. Visualizing 15 focal plasmablast clonal families in AIRRscape, we found five were common to both patient cohorts, often in multiple patients. The most common convergent lineage CF1, seen in both Parameswaran et al. [30] and Zanini et al. [31], was found not to be broadly neutralizing against all DENV serotypes [29]. However, the second most commonly found lineage, CF7, includes the mAb J9 that Durham et al. [29] found to be broadly neutralizing. That one of two tested convergent antibody lineages shows broad neutralization suggests further investigations of convergence may be beneficial for focusing antibody discovery and vaccine design. Broad neutralization could be particularly important for treating dengue, where antibody-enhanced disease is a problem.

As AIRRscape is in active development, we note some limitations and also envision future enhancements. AIRRscape is primarily an exploratory tool focused on visualization but various metrics could be used to formally measure overlap between repertoires [17,59]. AIRRscape does not visualize all of the sequences in the bulk BCR repertoire datasets; due to size and computing limitations in the Shiny application, large datasets are collapsed when V-gene + J-gene assignments as well as CDR3 amino acid motifs are identical. Similarly, to facilitate construction of topologies of CDR3 amino acid motifs we focus on those with exact same lengths, although it has been shown that functionally similar antibodies may have different CDR3 lengths [60,61]. While the variability of CDR3 is typically crucial for defining antigen specificity [49,62], other CDRs can also be important in antigen recognition; in those cases, AIRRscape could be modified to visualize changes outside of CDR3. With respect to identifying convergence, AIRRscape creates bins based on assigned germline V-gene families; thus, CDR3 topologies can indicate convergence even if antibodies have different germline V-genes within a family. While this definition of convergence is intended to cluster closely related germline genes such as IGHV3-53 & IGHV3-66, it may be too broad in some cases. As an aid, we include the full V-gene assignments in the topology tip names, and we note that restricting the antibody list to one gene will limit searches to only convergences with matching V-genes. Also, while AIRRscape is focused on amino acid motifs, we acknowledge that amino acid sequence convergence is an imperfect predictor of functional convergence. Lastly, although AIRRscape can quickly and easily search for known antibody sequences, examine overall repertoire patterns, and find convergence, users currently cannot initiate searches for identical motifs across datasets, although new tools with this functionality are being developed [63].

In summary, AIRRscape is a useful tool for quick and thorough exploration of large BCR repertoire datasets. Moreover, AIRRscape is novel among the suite of publicly available tools in visualizing convergence of antibody motifs. We have released the source code on GitHub so that interested research groups may use other datasets or incorporate more features. While the datasets from this study are pre-loaded, the tool allows users to visualize combinations of other AIRR datasets following AIRR Community standards. The code could be further modified to meet the desires of immuno-informaticians, virologists, and immunologists, for instance with

respect to the CDR3 sequence identity thresholds, phylogenetic tree options, or color and visualization schemes. While our focus is on infectious disease, AIRRscape could be used for autoimmunity and cancer biology, for example to compare healthy vs. autoimmune or cancer patient repertoires. With minor code modifications, AIRRscape could also be used to visualize TCR repertoires and their CDR3 motifs. We hope that AIRRscape will be useful for the research community, and we encourage interested parties to contribute additional features that would further enable identification of convergent responses.

## Supporting information

**S1 Data. Supplementary figures. Fig A in S1 Data. SARS-CoV-2 convergent clonotypes to mAb C102 in the 3_4_11 bin.** An 80% identity threshold is used to calculate convergence. Tips are colored by dataset source. Purple tips are published anti-COVID-19 antibodies from 12 different studies, dark gray tips are antibody sequences from a healthy donor BCR repertoire, and orange through brown shaded tips are antibody sequences from COVID-19 patient BCR repertoires. **Fig B in S1 Data. SARS-CoV-2 convergent clonotypes to mAb C125 in the 1_3_16 bin**. An 80% identity threshold is used to calculate convergence. Tips are colored by dataset source. Purple tips are published anti-COVID-19 antibodies from 7 different studies, dark gray tips are antibody sequences from a healthy donor BCR repertoire, and orange through brown shaded tips are antibody sequences from COVID-19 patient BCR repertoires. **Fig C in S1 Data. SARS-CoV-2 convergent clonotypes to mAb BD-494 in the 3_6_11 bin.** An 80% identity threshold is used to calculate convergence. Tips are colored by dataset source. Purple tips are published anti-COVID-19 antibodies from 15 different studies, dark gray tips are antibody sequences from a healthy donor BCR repertoire, and orange through brown shaded tips are antibody sequences from COVID-19 patient BCR repertoires. **Fig D in S1 Data. AIRRscape heatmaps comparing anti-HIV-1 antibodies and bulk BCR repertoires of eight HIV-1 patients. Fig E in S1 Data. HIV-1 convergent clonotypes to mAb CAP351_6m_6041 (Setliff et al. 2018; Fig 3) in the 1_6_14 bin.** A 70% identity threshold is used to calculate convergence. Tips are colored by dataset source. Purple tips are published anti-COVID-19 antibodies, and green shaded tips are antibody sequences from HIV-1 patient BCR repertoires. **Fig F in S1 Data. AIRRscape heatmaps comparing isolated dengue plasmablasts and bulk BCR repertoires of dengue patients from Colombian and Nicaraguan cohorts. Fig G in S1 Data. Dengue convergent clonotypes to CF6 (Zanini et al. 2018).** An 80% identity threshold is used to calculate convergence. Tips are colored by dataset source. Purple tips are plasmablast sequences reported by Zanini et al. (2018) isolated from two Colombian patients (d13 and d20), blue tips are antibody sequences from the BCR repertoire of patient d13, and gold tips are antibody sequences from a cohort of Nicaraguan patient BCR repertoires. **Fig H in S1 Data. Dengue convergent clonotypes to CF7 (Zanini et al. 2018).** An 80% identity threshold is used to calculate convergence. Tips are colored by dataset source. Purple tips are plasmablast sequences reported by Zanini et al. (2018) isolated from two Colombian patients (d13 and d20), blue tips are antibody sequences from the BCR repertoire of patient d13, and gold tips are antibody sequences from a cohort of Nicaraguan patient BCR repertoires. **Fig I in S1 Data. Dengue convergent clonotypes to Parameswaran et al. (2018) motif ARQIGNWFDP similar to CF1 (Zanini et al. 2018).** An 80% identity threshold is used to calculate convergence. Tips are colored by dataset source. Purple tips are plasmablast sequences reported by Zanini et al. (2018) isolated from two Colombian patients (d13 and d20), blue tips are antibody sequences from the BCR repertoire of patient d13, and gold tips are antibody sequences from a cohort of Nicaraguan patient BCR repertoires. **Fig J in S1 Data. SARS-CoV-2, HIV-1, & dengue convergent clonotypes to anti-SARS-CoV-2 mAb C102 in**

**the 3_4_11 bin.** An 80% identity threshold is used to calculate convergence. Tips are colored by dataset source. Purple tips are published anti-COVID-19 antibodies from 12 different studies, dark gray tips are antibody sequences from a healthy donor BCR repertoire, and orange through brown shaded tips are antibody sequences from COVID-19 patient BCR repertoires. Green shaded tips are antibody sequences from HIV-1 patient BCR repertoires. Gold tips are antibody sequences from a cohort of Nicaraguan dengue patient BCR repertoires. **Fig K in S1 Data. SARS-CoV-2, HIV-1, & dengue convergent clonotypes to anti-SARS-CoV-2 mAb BD-494 in the 3_6_11 bin.** An 80% identity threshold is used to calculate convergence. Tips are colored by dataset source. Purple tips are published anti-COVID-19 antibodies from 15 different studies, dark gray tips are antibody sequences from a healthy donor BCR repertoire, and orange through brown shaded tips are antibody sequences from COVID-19 patient BCR repertoires. Green shaded tips are antibody sequences from HIV-1 patient BCR repertoires. Gold tips are antibody sequences from a cohort of Nicaraguan dengue patient BCR repertoires. **Fig L in S1 Data. HIV-1 & dengue convergent clonotypes to anti-HIV mAb 02-o (Setliff et al. 2018; Fig 4) in the 1_4_13 bin.** An 80% identity threshold is used to calculate convergence. Tips are colored by dataset source. Purple tips are published anti-HIV-1 antibodies, while green shaded tips are antibody sequences from HIV-1 patient BCR repertoires. Blue tips are antibody sequences from the BCR repertoire of dengue patient d13, and gold tips are antibody sequences from a cohort of Nicaraguan dengue patient BCR repertoires. **Fig M in S1 Data. SARS-CoV-2 & dengue convergent clonotypes to anti-dengue mAb CF6 (Zanini et al. 2018) in the 1_4_13 bin.** An 80% identity threshold is used to calculate convergence. Tips are colored by dataset source. Purple tips are plasmablast sequences reported by Zanini et al. (2018) isolated from two Colombian dengue patients (d13 and d20), blue tips are antibody sequences from the BCR repertoire of dengue patient d13, and gold tips are antibody sequences from a cohort of Nicaraguan dengue patient BCR repertoires. Brown shaded tips are antibody sequences from COVID-19 patient BCR repertoires.
(PDF)

## Acknowledgments

We would like to acknowledge members of the CZ Biohub Data Science platform for thoughtful discussion, and Sandra L. Schmid for comments on the manuscript. We would also like to thank Craig Kapfer and John Hanks for assistance with deploying AIRRscape onto the web.

## Author Contributions

**Conceptualization:** Eric Waltari, John E. Pak.

**Data curation:** Eric Waltari.

**Formal analysis:** Eric Waltari.

**Methodology:** Eric Waltari.

**Resources:** Krista M. McCutcheon, Joan Wong, John E. Pak.

**Software:** Eric Waltari, Saba Nafees.

**Supervision:** Krista M. McCutcheon, Joan Wong, John E. Pak.

**Validation:** Eric Waltari, Saba Nafees, Joan Wong, John E. Pak.

**Visualization:** Eric Waltari.

**Writing – original draft:** Eric Waltari.

**Writing – review & editing:** Eric Waltari, Saba Nafees, Krista M. McCutcheon, Joan Wong, John E. Pak.

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
