## [Decision Letter · Decision Letter 0]

18 Jun 2022

Dear Dr. Waltari,

Thank you very much for submitting your manuscript "AIRRscape: an interactive tool for exploring B-cell receptor repertoires and antibody responses" for consideration at PLOS Computational Biology.

As with all papers reviewed by the journal, your manuscript was reviewed by members of the editorial board and by several independent reviewers. In light of the reviews (below this email), we would like to invite the resubmission of a significantly-revised version that takes into account the reviewers' comments.

We cannot make any decision about publication until we have seen the revised manuscript and your response to the reviewers' comments. Your revised manuscript is also likely to be sent to reviewers for further evaluation.

Sincerely,

Gur Yaari

Guest Editor

PLOS Computational Biology

Rob De Boer

Deputy Editor

PLOS Computational Biology

Reviewer's Responses to Questions

**Comments to the Authors:**

Reviewer #1: Provided as an attachment

Reviewer #2: See attachment.

Reviewer #3: Review is uploaded as an attachment.

Reviewer #4: This manuscript describes AIRRscape, an R Shiny application that can be run in the browser, and that allows for the comparison of antibody sequences based on V and J gene, and CDR3 length. It visualizes these antibody characteristics in the form of heatmaps and can examine the convergence of clonotypes by constructing phylogenetic-like trees. The three use cases illustrate potential applications of the tool. The source code is publicly available on GitHub. With the major comments addressed, especially concerning how it compares to other similar tools, I think AIRRscape could be a useful contribution to the field, particularly for researchers with less programming experience.

Major comments:

- Please provide an overview table of what new or improved functionality AIRRscape provides compared to other existing tools for antibody analyses, such as VDJtools https://doi.org/10.1371/journal.pcbi.1004503, immunarch http://doi.org/10.5281/zenodo.3367200, PASA https://doi.org/10.1371/journal.pcbi.1008607.

- Please update Figure 1 to show an overview of all types of AIRR analyses that are possible in AIRRscape.

- How are motifs defined and how are they discovered (e.g., line 64)? In line 203, the authors mention “healthy control motifs” – what does this mean? Please include the definitions and explanations of motifs in the manuscript.

- In multiple places throughout the manuscript, the authors mention the increasing size of the antibody repertoires and how AIRRscape can help. For example, the author summary (line 27) states: “With AIRRscape, we enable large-scale immune repertoire visualization and analysis...”. How scalable is it? Please quantify scalability.

Minor comments:

- R and R studio versions are missing, along with dependencies. It would be nice to have all that resolved automatically if possible or to have a list of required packages and tools with versions. With this resolved, AIRRscape worked well locally for me.

- The authors mention “processed flat files” multiple times (e.g., line 142): what does flat refer to in this context?

- In line 172, it says that users may choose to construct the topology via either neighbor-joining or parsimony tree building methods. Is this performed using some package or is it a native AIRRscape implementation?

**Have the authors made all data and (if applicable) computational code underlying the findings in their manuscript fully available?**

Reviewer #1: Yes

Reviewer #2: Yes

Reviewer #3: Yes

Reviewer #4: Yes

PLOS authors have the option to publish the peer review history of their article (what does this mean?). If published, this will include your full peer review and any attached files.

Reviewer #1: No

Reviewer #2: **Yes: **Krishna M. Roskin

Reviewer #3: No

Reviewer #4: No
---

## [Decision Letter · Decision Letter 1]

4 Sep 2022

Dear Dr. Waltari,

We are pleased to inform you that your manuscript 'AIRRscape: an interactive tool for exploring B-cell receptor repertoires and antibody responses' has been provisionally accepted for publication in PLOS Computational Biology.

Before your manuscript can be formally accepted you will need to complete some formatting changes, which you will receive in a follow up email. A member of our team will be in touch with a set of requests. Also, please address the minor commnets raised by reviewer #4 (see below).

Best regards,

Gur Yaari

Guest Editor

PLOS Computational Biology

Rob De Boer

Section Editor

PLOS Computational Biology

Reviewer's Responses to Questions

**Comments to the Authors:**

Reviewer #1: The points raised in my review have been addressed. Improvements to the scripts and provision of a web-based facility significantly improve usability for the average user.

Reviewer #3: None. Authors have responded to all comments.

Reviewer #4: The authors have successfully addressed the comments I had.

Some very minor points:

- Updating the added Table 1 to emphasize the main advantages of AIRRscape might make it clearer to readers why they should use AIRRscape and not some of the other tools listed, for a specific analysis, e.g., easier to use, visualization options.

- While this information is now available in README under the Tips heading, having a very short description or table title for the two tables in the AIRRscape tab might make it more user-friendly.

**Have the authors made all data and (if applicable) computational code underlying the findings in their manuscript fully available?**

Reviewer #1: Yes

Reviewer #3: Yes

Reviewer #4: Yes

PLOS authors have the option to publish the peer review history of their article (what does this mean?). If published, this will include your full peer review and any attached files.

Reviewer #1: No

Reviewer #3: No

Reviewer #4: No

---

## [Editor Report · Acceptance letter]

15 Sep 2022

PCOMPBIOL-D-22-00462R1 

AIRRscape: an interactive tool for exploring B-cell receptor repertoires and antibody responses

Dear Dr Waltari,

I am pleased to inform you that your manuscript has been formally accepted for publication in PLOS Computational Biology. Your manuscript is now with our production department and you will be notified of the publication date in due course.

With kind regards,

Olena Szabo
